# Surface Modification of Pure Magnesium Mesh for Guided Bone Regeneration: In Vivo Evaluation of Rat Calvarial Defect

**DOI:** 10.3390/ma12172684

**Published:** 2019-08-22

**Authors:** Shuang Wu, Yong-Seok Jang, Yu-Kyoung Kim, Seo-Young Kim, Seung-O Ko, Min-Ho Lee

**Affiliations:** 1Department of Dental Biomaterials, Institute of Oral Bioscience and Institute of Biodegradable Material, BK21 plus Program, School of Dentistry, Chonbuk National University, Jeonju 54896, Korea; 2Department of Oral and Maxillofacial Surgery, Institute of Oral Bioscience, BK21 plus Program, School of Dentistry, Chonbuk National University, Jeonju 54896, Korea; 3Research Institute of Clinical Medicine of Chonbuk National University, Biomedical Research Institute of Chonbuk National University Hospital, Jeonju 54896, Korea

**Keywords:** guided bone regeneration, magnesium mesh, plasma electrolytic oxidation, hydrothermal treatment, rat calvarial defect model

## Abstract

Guided bone regeneration is a therapeutic method that uses a barrier membrane to provide space available for new bone formation at sites with insufficient bone volume. Magnesium with excellent biocompatibility and mechanical properties has been considered as a promising biodegradable material for guided bone regeneration; however, the rapid degradation rate in the physiological environment is a problem to be solved. In this study, surface modification of pure magnesium mesh was conducted by plasma electrolytic oxidation and hydrothermal treatment to form a densely protective layer on the Mg substrate. The protective layer mainly consisted of Mg(OH)_2_ with the amorphous calcium phosphate. Then, weight loss measurement and Micro-CT imaging were performed after an immersion test in a simulated body fluid. The effect of surface modification of the magnesium mesh on the guided bone regeneration was evaluated through an in vivo test using the rat calvarial defect model. The biodegradation of the magnesium mesh was identified to be significantly retarded. Additionally, the surface modification of Mg also can improve the bone volume and bone density of calvarial defect in comparison with that of the pristine Mg mesh.

## 1. Introduction

As sufficient bone volume is required for a successful implant treatment in the orthopedic and prosthodontics fields, the guided bone regeneration (GBR) has been conducted in patients with inadequate bone volume or serious bone defects [1]. Particularly in the surgery of maxillofacial, GBR using the membrane or mesh has been employed to increase the volume of the alveolar bone and bone regeneration at a bony window or bony defect [2,3]. To perform GBR successfully, the materials used must have excellent biocompatibility, tissue barrier function, host tissue integration, clinical manageability, and space maintenance [4,5,6,7]. Various bioresorbable or non-resorbable materials are conventionally used in GBR. Titanium [8] and stainless steel [9,10], which are non-resorbable metallic materials with excellent strength and ductility, can prevent the material collapse of wide bony defects and easily form a contour according to the shape of the defect. However, the mismatch in elastic modulus between the material and natural bone can produce a stress-shielding effect, and mastication and stimulation of the oral mucosa during long-term implantation can induce an inflammatory response and mesh exposure [11]. Thus, secondary surgery involving the removal of the bio-inert mesh is required to prevent these side effects, which can leave patients with a financial and temporal burden, as well as secondary tissue damage and physical distress [7]. For these reasons, the usage of bioresorbable membranes in GBR has been increasing owing to their good biocompatibility and gradual resorption by the body. However, bioresorbable polymeric membranes can get damaged by external force owing to the lower strength and collapse by structural faults resulting from the degradation. Based on circumstances, non-resorbable metallic meshes were applied at a wide bony defect and resorbable polymeric membranes were applied at a region where a relatively weak force was loaded. Recently, studies of a bioabsorbable magnesium (Mg) mesh for GBR have been actively pursued to strengthen the advantage and supplement the weakness of bioresorbable and non-resorbable membranes [1].

Magnesium and its alloys have attracted much attention from researchers as a promising biodegradable material for GBR, because they have enough rigidity to maintain the space at the site, forming a new bone for a long time in comparison with conventional bioresorbable polymeric membranes [12]. Magnesium is an essential mineral element for human bone formation and metabolism; meanwhile, any remaining excess can be excreted by urine [13,14,15]. In addition, the mechanical properties of magnesium, such as the density, modulus of elasticity, and compressive yield strength, are similar to those of the human cortical bone, which can minimize the stress shielding effect [14,16]. Although magnesium has the above advantages, low corrosion resistance is a major drawback to use in implantable medical devices. The evolution of gas and alkalization, which is caused by corrosion of the magnesium, can cause inflammation and hinder bone formation [17]. Therefore, controlling the initial degradation rate after implantation to prevent these side effects and maintain sufficient mechanical strength until the damaged tissue is completely healed is important.

Various surface modification methods have been developed to enhance biocompatibility and retard the biodegradation rate of magnesium medical devices. Calcium-phosphate coating as a biomimetic coating is one of the most effective surface modification methods to increase biocompatibility and control the initial degradation of magnesium implants by reducing exposure to the corrosive environment [18,19,20,21]. In addition, plasma electrolytic oxidation (PEO) as an electrochemical technique is a preferable surface modification method to retard the corrosion rate by the formation of porous protective magnesium oxide layer films [22]. Recently, many PEO methods for magnesium alloys have been developed using various conditions of applied power and electrolytes with different compositions and concentrations [22,23,24,25,26]. For instance, Seyfoori et al. [22] and Durdu and Usta [26] identified that the corrosion resistance of magnesium was significantly improved by the formation of a thick and dense layer on the surface through PEO in the phosphate-based electrolyte. The hydrothermal treatment (HT) has been reported as a cost-effective method to form an anti-corrosion coating layer on the Mg surface, which can improve the corrosion resistance significantly [27]. Many previous studies have reported that the combination of PEO and HT could enhance biocompatibility and increase the corrosion resistance of Mg more than they would if they were conducted separately [28,29].

The current aim of this study was to evaluate the potential of the surface-modified Mg as a biomedical material. The degradation of magnesium was investigated by the weight loss measurement and Micro-CT imaging after an immersion test in a simulated body fluid. The osteogenesis of magnesium was evaluated by an in vivo test in an eight mm-critical size of the rat calvarial model.

## 2. Materials and Methods

### 2.1. Preparation of Magnesium Mesh

Magnesium meshes with a mesh diameter of 10 mm and hole diameter of 0.4 mm were manufactured using commercially available 99.9% pure magnesium foils with dimensions of 100 × 100 × 0.1 mm (as rolled, GoodFellow, Huntingdon, UK) through laser micro-processing, as shown in Figure 1A. A rectangular lug protruding from the mesh was created for use as an electrical connection during PEO and removed after specimen fabrication.

### 2.2. Surface Treatment

Corrosion products formed on the surface of magnesium meshes during laser micro-processing were removed by chemical cleaning in a solution consisting of 200 g of chromium trioxide (CrO_3_), 10 g of silver nitrate (AgNO_3_), and 20 g of barium nitrate (Ba(NO_3_)_2_) in 1 L distilled water for 1 min, according to ASTM G1-03 [30]. Then, the specimens were cleaned by distilled water and absolute ethanol, and dried in a stream of dry air. 

To conduct the PEO treatment, the magnesium meshes and platinum (Pt) foil were connected to the anode and cathode of the constant DC power (Kwangduck FA, Chilgok-gun, Korea), respectively. The PEO process was carried out in an electrolyte, which consisted of 0.1-M sodium hydroxide (NaOH), 0.1 M glycerol, and 0.1-M sodium phosphate (Na_3_PO_4_) at a current density of 300 mA/cm^2^ for 1 min.

The hydrothermal treatment was conducted by immersing the PEO-treated meshes in the solution of 0.5 M sodium hydroxide (NaOH) and 0.2 M Ca-EDTA (C_10_H_12_CaN_2_Na_2_O_8_) at 90 °C in a high-temperature oven for 24 h. Subsequently, the specimens were washed by distilled water and dried at 40 °C in a dry oven for 24 h.

### 2.3. Surface Characterization

To observe the microstructure and the chemical composition of specimens, the untreated Mg mesh, PEO-treated Mg mesh and PEO/HT-treated Mg mesh were identified through the scanning electron microscopy microscope (SEM; JSM-5900, JEOL, Tokyo, Japan) in conjunction with the energy dispersive spectroscopy (EDS; Bruker, Billerica, MA, USA). The cross-sectional analysis for the coating layer of the PEO-treated Mg mesh and PEO/HT-treated Mg mesh was conducted using the field emission scanning electron microscopy (FE-SEM, SU-70; Hitachi, Tokyo, Japan) after ion milling. Phase identification of the untreated Mg mesh, PEO-treated Mg mesh and PEO/HT-treated Mg mesh were analyzed using the X-ray diffractometer (X’pert Powder, PANalytical, Almelo, The Netherlands) with 2θ ranging from 10° to 90° by Cu-Kɑ (λ = 0.154060 nm) at a step of 0.03°. In all of the above tests, three samples per group were used.

### 2.4. Immersion Test

The immersion test was conducted in the simulated body fluid (SBF) solution for eight weeks to confirm the effects of the surface treatment on the corrosion behavior of the Mg meshes. The SBF solution for the immersion test was fabricated by adding 0.185 g/L of CaCl_2_·2H_2_O, 0.35 g/L of NaHCO_3_ and 0.09767 g/L of MgSO_4_ in Hank’s balanced salt solution (HBSS; H2387, Sigma Chemical Co., St. Louis, MO, USA) according to the manufacturer’s instructions, and the pH was adjusted to 7.4. five meshes per group, which were the untreated Mg mesh, PEO treated Mg mesh, and PEO/HT treated Mg mesh, were immersed in a plastic tube containing 50-mL of the SBF solution. Then, an immersion test was carried out for eight weeks at 37 °C in an incubator. SBF solutions were refreshed once a week to minimize the effect of pH and the variation of ions concentration in SBF solutions. The morphological changes of meshes during the immersion test was observed using Micro-CT (Skyscan 1076, Skyscan, Aartselaar, Belgium) at four weeks and eight weeks. The weight of the meshes for each group after the eight-week immersion test were measured after the corrosion product was removed using the solution for chemical cleaning, and then the weight loss was calculated.

### 2.5. Animal Experiment

All animal experiments were conducted under ethical clearance, which was authorized by the Institutional Animal Care and Use Committee of the Chonbuk National University Laboratory Animal Center, Jeonju, Korea (Approved number: CBNU 2018-004).

Prior to the experiment, the rats were fed in plastic cages in an animal room with a constant temperature and humidity for one week to adapt to the environment. Sixteen male Sprague-Dawley rats (eight weeks old and body weight of 230 ± 20 g) were randomly divided into two groups for the untreated magnesium mesh (UT group) and HT-treated magnesium mesh after PEO (PEO/HT group). Eight rats in each group were used, and four rats in each group were sacrificed at two time-points: The fourth week and the eighth week.

#### 2.5.1. Surgical Procedures

All surgery was performed under aseptic conditions. Rats were anesthetized by an intramuscular injection of 50 mg/kg of Zoletil (Zoletil 50, Virbac Laboratories, France) and 15 mg/kg of Xylazine hydrochloride (Rompun, Bayer, Korea). A surgical position (calvaria) was shaved with the electric clipper, then disinfected with iodine scrubs, and 0.5 mL of 1% Lidocaine with epinephrine (1:100,000) was administered at the surgical site to reduce bleeding. For the operation, a two-cm midsagittal incision was made. Then, the periosteum was bluntly dissected to expose the surgical site. An eight-mm-diameter defect was made by the trephine bur with an endodontic motor (X-SMART, Dentsply, Tokyo, Japan) under saline irrigation. With the calvarium around one-mm thick, care was taken not to damage the dura or brain. The margin of the defect was washed to remove the residual bone fragments. The bone defect was completely covered using the magnesium mesh, and the periosteum that shielded the magnesium mesh and was sutured with a bioabsorbable silk (5-0 poly-glactin 910, Vicryl, Ethucon, UK). Then, the skin flaps were sutured by a 4/0 silk (4/0 black silk, Ailee, Korea), as shown in Figure 1B. After anabiosis, rats were housed in separate cages and were fed ad libitum. The antibiotics (amikacin, Sanu media, Yesan-gun, Korea) were administrated at zero, 24, and 48 h for postoperative anti-infection. The rats were sacrificed by euthanasia using an overdose of the thiopental sodium (ChoongWae Pharma Corporation, Seoul, Korea) at the fourth and the eighth week. The excised blocks, including the membrane, soft tissue, and calvaria were fixated in a 10% buffered formalin solution for 1 d.

#### 2.5.2. Micro-computed Tomography Scanning (Micro-CT)

The dissected block specimens were immersed into 10% formalin in a plastic tube with a label, then sent to the Center for University-Wide Research Facilities (CURF) at Chonbuk National University at the four-week and eight-week time points. Dissected specimens were scanned using Micro-CT (Skyscan 1076, Skyscan, Aartselaar, Belgium) at 100 kV and 100 μA with the 680 ms of exposure time, 0.6° of rotation intervals, and 360° of rotation angle, to obtain qualitative and quantitative data of bone regeneration. The 3D images reconstructed with the NRecon software (Ver 1.7.0, Skyscan, Aartselaar, Belgium).

For the calvarial defect, a cylindrical region of interest (ROI) with a diameter of 8 mm (the size of the cranial defect) and height about 3 mm (including the newly formed bone and magnesium mesh), was chosen for calculating the bone qualitative and quantitative results. The gray thresholds of the osseous tissue were 110–255 in Hounsfield units (HU). The new bone volume (mm^3^) and new bone mineral density (g/mm^3^) were quantified using the CTAn software (Skyscan, Aartselaar, Belgium). The 3D structure was reconstructed by the ^®^CTvoxs software (Ver 3.0, Skyscan, Aartselaar, Belgium).

#### 2.5.3. Histological Analysis and Fluorescent Analysis

When Micro-CT scanning was finished, the specimen began a series of processes for embedding and staining, following the methodology used in our previous study [31]. Four rats per group were sacrificed for the histological analysis and fluorescent analysis, respectively, at the fourth and eighth weeks. 

After Micro-CT scanning, the blocks were fixed in a new 10% formalin for 1 d and stained in the Villanueva solution (Polysciences, Inc., Eppelheim, Germany). Villanueva is a bone stain which is widely used for effective staining of the mineralized or uncalcified bone, especially for the study of new bone formation in hydroxyapatite implants and bone grafts. Then, the blocks were dehydrated in ethanol with varying concentrations (80%, 90%, 95%, and 100%) and 100% acetone. To embed the blocks in resin, the blocks were pre-permeated with methylmethacrylate (MMA monomer, Yaruki Pure Chemicals Co., Ltd., Kyoto, Japan) under a vacuum for 2 h and were then infiltrated with the polymerization mixture (MMA solution adding 2 wt.% benzoyl peroxide) at 35 °C for 3 d and at 60 °C for 1 d. Prepared resin blocks were cut to slices with an 0.8-mm thickness through the center line of the defect, using a low-speed saw (EXAKT 300 CP, EXAKT Technologies Inc., Norderstedt, Germany). Slices were grinded to a 40-µm thickness for the histological analysis using a micro-grinding system (EXAKT 400 CS, EXAKT Technologies Inc., Norderstedt, Germany). The histomorphometric analysis was conducted by the optical microscopy (x8; EZ4D, Leica and x50; DM2500, Leica, Wetzlar, Germany).

Sequential double fluorochrome labelling was conducted to identify the dynamics of the new bone formation in the defect beneath the pure magnesium mesh. The red-alizarin complexion (1.67 mg/kg) and green-calcein (1.25 mg/kg) (both from Sigma, St. Louis, MO, USA) were subcutaneously administered at different times. For the four-week sacrificed rats, the alizarin was injected after surgery immediately, and the calcein was injected at two weeks post-operation. For the eight-week sacrificed rats, the alizarin and calcein were sequence-injected at four-week and six-week postoperative times. The specimens were prepared in the same way as the histological specimens and evaluated with a confocal laser scanning microscope (CLSM 510 Meta, Zeiss, Jena, Germany).

#### 2.5.4. Statistical Analyses

The data were presented as mean ± standard deviation (SD). The one-way ANOVA with the post hoc Tukey test was performed, and P < 0.05 was considered statistically significant.

## 3. Results

### 3.1. Surface Characterization

Figure 2A–C shows the optical images of the Mg meshes before and after the surface treatment. The color of the surface was changed from a Mg intrinsic silver-white to gray after the PEO and to a bright yellowish brown after the HT treatment. The porous structure including about 8% phosphorus (P) with 1–2 μm micro-pores was observed on the surface of the PEO-treated Mg mesh. As the cross-sectional SEM image in Figure 2H shows, the average PEO coating thickness was 1.07 ± 0.19 μm, which consisted of a dense bottom layer with a thickness of 0.8 μm and a porous top layer with a thickness of 0.2 μm. After the HT treatment, the plate-like nanocrystals including 2.8% P and 1.7% calcium (Ca) were densely formed on the PEO coating layer, which sealed the pores of the PEO coating perfectly, as shown in Figure 2F,I. The thickness of the PEO/HT coating was approximately 9.68 ± 1.05 μm, and a bottom layer approximately 2 μm denser than the upper layer was observed. The main crystal structures of PEO and PEO/HT were identified as the magnesium oxide (MgO) and magnesium hydroxide (Mg(OH)_2_), respectively, in the results of the XRD analysis in Figure 3.

### 3.2. Changing of Mg-mesh in Immersion Test

Figure 4A shows the Micro-CT images of the Mg meshes degraded after immersion in SBF for four and eight weeks. Untreated Mg meshes were conspicuously degraded as compared with PEO- and PEO/HT-treated Mg meshes; however, the PEO-treated Mg meshes were degraded in a very small area, and the PEO/HT-treated meshes were barely degraded after the four-week immersion. A little degradation of the Mg meshes were observed in the PEO/HT treated meshes, and the most severe degradation occurred in the untreated Mg meshes after immersion for eight weeks. When compared with the weight loss after removing the corrosion products of the Mg meshes immersed for eight weeks, the weight loss ratio of the untreated mesh was 39.36% ± 3.64%, which was significantly higher than that of the PEO- and PEO/HT- treated meshes. Among the surface treated samples, the weight loss ratio (21.67% ± 4.56%) of the PEO mesh also had a significant difference with that (15.08% ± 2.89 %) of the PEO/HT mesh. As mentioned above, the surface-modified PEO/HT group significantly improved the biocorrosion resistance of the pure Mg. 

### 3.3. Degradation of Mg Meshes and New Bone Formation in vivo

To confirm the effect of the surface treatment on the biodegradation and bone formation of the Mg mesh, the untreated Mg mesh and PEO/HT-treated Mg mesh were used in the in vivo experiment. Figure 5 shows the representative three-dimensional reconstruction images of the Micro-CT data for the degraded Mg meshes, and a new bone formed inside the rat calvarial defects. In terms of the degradation of the Mg meshes, the untreated Mg mesh had degraded more rapidly than the PEO/HT-treated Mg mesh at four weeks. At eight weeks post-implantation, the untreated Mg mesh was completely degraded beyond recognition; however, the PEO/HT-treated Mg mesh remained in perfection except for a localized small area, which was consistent with the results of the immersion test. It was identified that new bones were grown beneath the mesh toward the center from the edge of the calvarial defect in both groups. As shown in Figure 6, the volume and mineral density of the new bone that formed on the surfaces of both Mg meshes gradually increased during healing, and those values of the PEO/HT mesh were higher than that of the untreated Mg mesh at both four and eight weeks. Although there was no statistically significant difference between the groups, both the bone volume and mineral density of the PEO/HT mesh were higher than that of the untreated Mg meshes at eight weeks.

Figure 7 represents the histological images of the newly formed bone in rat calvarial defects of untreated and PEO/HT-treated Mg mesh groups. The new formation of the bone tissue in both groups was primarily derived from the margin of the defect, then grew toward the central part along the Mg meshes without the collapse of the new bone or Mg mesh. At four weeks, it was observed that the new bone was separated from the untreated Mg mesh by gas bubbles generated by corrosion of the Mg mesh; meanwhile, the new bone formed beneath the PEO/HT-treated Mg mesh was grown in contact with the mesh. At eight weeks, the untreated Mg mesh was partially degraded; however, the PEO/HT-treated Mg mesh remained in its original shape excluding slight corrosion around the hole in the mesh. In addition, gas pockets were no longer observed on the untreated Mg mesh at eight weeks, but a small gas bubble was observed in the new bone formed on the PEO/HT mesh at eight weeks.

To clarify the new bone formation dynamics in the defect beneath the magnesium mesh, the sequential fluorochrome labels were administered subcutaneously for each group at different times. The sequence of the new bone formation was confirmed by the red label of the alizarin complexone and the green label of calcein. After administration, fluorochrome labels, bound to Ca ions of the body, were incorporated in the form of HA at the sites of mineralization. This means that the fluorescent label delimited the mineralization front at the time of administration. As a result, bone formation can be tracked by administrating different labels at specific time intervals. Figure 8 shows the results of the fluorescence test using the confocal laser scanning microscopy. Rats sacrificed at four weeks received the red-alizarin post-surgery and the calcein in two weeks. According to the working mechanism of the fluorochrome label, the green label of the calcein appeared inside the red label of the alizarin or two labels overlapped in both groups, which indicates that the deformation/remodeling of the new bone layer occurred. It was also observed that the gas pocket, which is highlighted by the blue line existed between the untreated Mg mesh and new bone. Meanwhile, the new bone layer formed on the PEO/HT-treated mesh was closely grown under the mesh. In the case of rats sacrificed at eight weeks after administering the alizarin and the calcein at four and six weeks, the position of the red and green label was clearly distinguished in contrast with the four-week group. In both the untreated and PEO/HT groups, the expression of the red label was observed in the new bone close to the mesh, and the green label was shown outside of the red label, which means that the growth of the new bone proceeded from beneath the mesh outwards.

## 4. Discussion

Bioabsorbable barrier membranes are widely using in the GBR of dentistry because the secondary surgery for removal of the membrane is not required. However, one of the shortcomings of absorbable membranes is the lack of rigidity, leading to the collapse into the wound and subsequent compromised volume or failure of bone regeneration. Even if successful in bone regeneration, the morphology often typically results in a rounded crestal ridge form, requiring extra grafting at the time of implantology [32,33]. The pristine Mg as a biomedical material rapidly degraded in the physiological environment, losing advantages of the metal rigidity and space maintenance, indicating its inability to provide the necessary support until the natural bone tissue completely heals. Meanwhile, the high corrosion rate of the magnesium or its alloys at the preliminary stage of implantation resulted in a gas accumulation and ions released as well as the variable pH at the tissue around the implant, which can cause a turbulence of the stable environment, interference with cell adhesion, and hindrance of bone formation [34,35]. Hence, a large number of researches have been performed to enhance the corrosion resistance and decrease the initial biodegraded rate of magnesium in a biomedical application. In this study, a protective coating containing Ca and P was formed on the pure magnesium mesh via a two-step method by the combination of PEO and HT treatments to retard the biodegradation of the Mg mesh, and its effects on biodegradation and bone regeneration were verified by the immersion test and in vivo tests for the GBR application. It was identified that the biodegradation rate of the Mg mesh significantly decreased, and bone regeneration improved by the PEO/HT treatment of the surface modification, compared with the pristine Mg mesh.

The PEO is a preferable method to retard the corrosion of magnesium and its alloys by forming a porous oxide film, because the thickness and surface morphology can be easily controlled by the change in the applied electrical power, processing time, and electrolyte composition [36,37]. In this study, a porous MgO layer containing approximately 8% P with a thickness of approximately 1 μm was uniformly produced on the surface of the pure Mg mesh by the PEO process. Although it was confirmed that the corrosion resistance of Mg improved by the PEO, the pores that exist in the PEO film results in a localized corrosion of the Mg substrate beneath the PEO film by the penetration of body fluid, including an aggressive chloride ion (Cl^−^) into the pores [38]. In addition, the calcium phosphate coating on the Mg implant for orthopedic usage can improve bone healing by an increase in the biocompatibility and corrosion resistance [39,40]. Thus, it is beneficial and cost-effective to improve the biocompatibility and retard the biodegradation by the composite surface treatment using the PEO and calcium phosphate coating. Many researchers have tried to seal the pores by the hydrothermal treatment using various aqueous solutions, including Ca and P ions, to enhance the corrosion resistance and biocompatibility. The pores are usually sealed by formation of the biofriendly calcium phosphate crystals and Mg(OH)_2_, based on reactions such as MgO + H_2_O → Mg(OH)_2_, Mg + 2H_2_O → Mg(OH)_2_, during the hydrothermal treatment [41]. In a previous study, when the protective calcium phosphate layer was formed on the AZ31 magnesium alloy by the hydrothermal treatment in an aqueous solution, including Ca-EDTA at 90 °C for 24 h after the formation of the PEO coating including P, it was identified that the corrosion resistance and bioactivity of the magnesium alloy had improved [42]. In another study [31], it was confirmed that P in the PEO coating and Ca in the electrodeposited layer formed on a Mg-Al-Zn-Ca alloy were chemically incorporated by the hydrothermal treatment, which increased the corrosion resistance and promoted the bone formation in vivo. In the present study, Mg(OH)_2_ was the main crystalline phase forming on the PEO/HT treated Mg mesh, which retarded the degradation of the Mg mesh significantly and enhanced the new bone formation, as confirmed by the immersion test and in vivo test. Janning et al. [43] proved that the bone growth enhancement around the slowly dissolving Mg(OH)_2_ cylinder in the rabbit model was caused by enhancing the osteoblast activity and decreasing bone resorption. According to the EDS and XRD data of the stated experiment, the Ca/P ratio of the specimens was less than 1.0, which suggests that the precipitated calcium phosphate is amorphous. However, this didn’t imply the absence of the microcrystalline phase. The mineral phase of the hard tissue has an apatite structure consisting of a non-stoichiometric HAP and trace element like Mg^2+^, CO_3_^2−^, Cl^−^ [44,45]. When the Mg concentration is high, the apatite exists in the form of (Ca_1–x_Mg_x_)_10_(PO_4_)_6_OH_2_, which has biocompatibility to the MG63 cells. These are consistent with the presented results of the animal test, the protective coating layer containing Ca-P of PEO/HT has a good affinity of the bone-implant interface, high corrosion resistance and an increase in the bone formation at the initial implantation.

The effect of the PEO/HT treatment on the degradation behavior and bone regeneration of the Mg mesh was studied by the in vivo test using the critical-size defect model of rat calvaria in this study. A calvarial defect of 8 mm has been generally used in animal testing to confirm GBR, as this critical size of 8 mm is considered to be the smallest size in which the bone cannot completely heal inherently without other guided material, and the calvarial has the benefit of poor blood supply and no need of extra fixation for stabilization [46,47,48]. In the results of the histological and bony formation dynamic analyses, as shown in Figure 7 and Figure 8, it was observed that the untreated Mg mesh was severely degraded, and several gas pockets were found between the Mg mesh and new bone at four weeks after implantation. Fast degradation and formation of gas pockets of the untreated Mg mesh hindered the initial cortical bone healing process and induced bone resorption, as the increased inner pressure in the gas pockets pushed the surrounding tissue away and disrupted the supply of blood and body fluid needed for the bone regeneration. Meanwhile, the PEO/HT-treated Mg mesh was rarely degraded, and no gas pocket was observed at four weeks due to the retardation of the initial corrosion of the Mg mesh by the dense PEO/HT coating. Furthermore, more new bone was grown along the surface of the mesh with good contact from the original old bone edge to the center of the defect. At eight weeks, most of the untreated mesh was degraded, and the cavities of the gas were not observed anymore. The diminishment of the gas generation might be due to the decrease in the corrosion rate of the mesh by the formation of byproducts, such as Mg(OH)_2_ and Mg_x_Ca_y_(PO4)_z_(OH)_n_ complex compounds, and the reduction in the surface area where gas can evolve. The disappearance of gas pockets indicated that the generation rate of H_2_ falls below the diffusion rate of H_2_ [49]. However, tiny gas pockets were observed in the new bone formed at the center of the PEO/HT mesh, which resulted from the localized corrosion of the mesh by the destruction of the PEO/HT coating. It was identified that the main component of the gas released by the Mg was hydrogen, and CO_2_ and CO constituted the residual gas [50]. When the Mg was placed in the body fluid environment, the hydrogen released rapidly (Mg + 2H_2_O → Mg(OH)_2_ + H_2_).

The limitation of this study is that complete bone healing in the critical size defect of rat calvaria was not confirmed at eight weeks in neither the untreated nor PEO/HT-treated mesh groups, which means that the experimental period should be extended in future study to identify the final duration for the completed bone regeneration by Mg.

## 5. Conclusions

In this study, the Mg(OH)_2_ layer including Ca and P was densely formed on the surface of the Mg mesh by a two-step treatment combining the plasma electrolytic oxidation and hydrothermal treatment. It was confirmed that the biodegradation of the Mg mesh was retarded by the surface treatment in a physiological environment, which could improve bone regeneration in the calvarial defect of rats.

## Figures and Tables

**Figure 1 materials-12-02684-f001:**
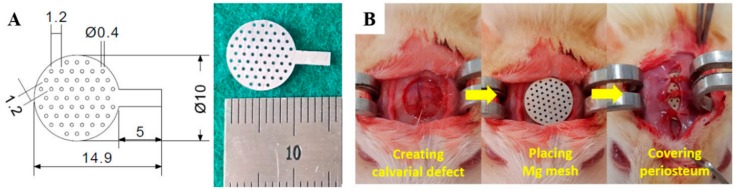
(**A**) Schematic diagram and optical image of the magnesium mesh used for the experiment; (**B**) operative procedure for the rat calvarial defect model.

**Figure 2 materials-12-02684-f002:**
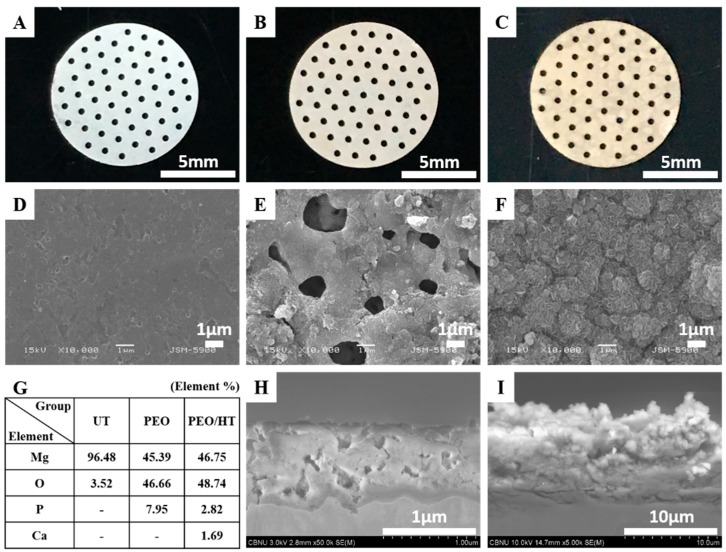
(**A**) Optical images of the untreated Mg mesh, (**B**) PEO-treated Mg mesh, (**C**) PEO/hydrothermal treatment (HT)-treated Mg mesh, (**D**) SEM surface morphologies of the untreated Mg mesh, (**E**) PEO-treated Mg mesh, (**F**) PEO/HT-treated Mg mesh, (**G**) chemical composition on the surfaces, (**H**) cross-sectional images of the PEO-treated Mg mesh, and (**I**) PEO/HT-treated Mg mesh.

**Figure 3 materials-12-02684-f003:**
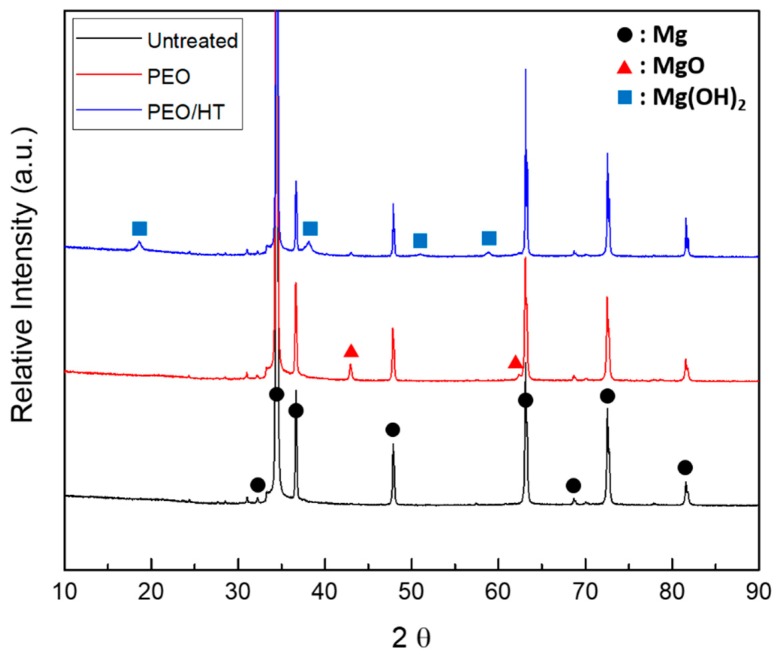
XRD patterns of the untreated Mg mesh, PEO-treated Mg mesh, and PEO/HT-treated Mg mesh.

**Figure 4 materials-12-02684-f004:**
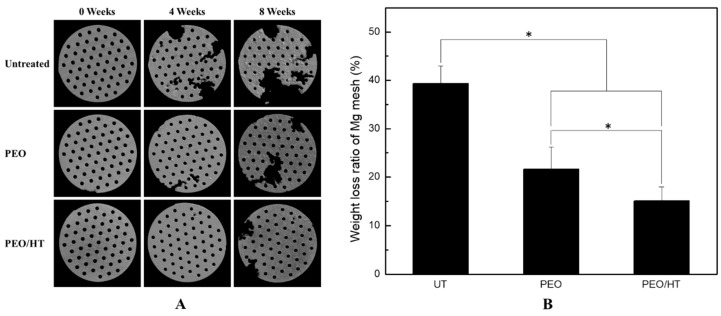
(**A**) 3D reconstruction for the Mg meshes after immersion in SBF for four and eight weeks, and (**B**) the weight loss ratio of the Mg meshes after immersion in SBF for eight weeks. (* means *p* < 0.05).

**Figure 5 materials-12-02684-f005:**
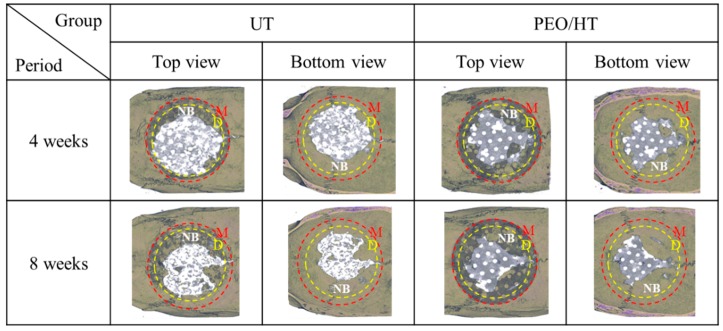
Micro-CT images of the degraded Mg meshes and new bone formed into rat calvarial defects after in vivo tests at four weeks and eight weeks. The yellow-dotted line signifies the area of the calvarial defect formed initially in surgery, demarcating the internal new bone and external old bone. The red-dotted line indicates the outer line of the Mg mesh. (Yellow circle D = defect; red cycle M = magnesium mesh; NB represented the newly formed bone inside the yellow circle).

**Figure 6 materials-12-02684-f006:**
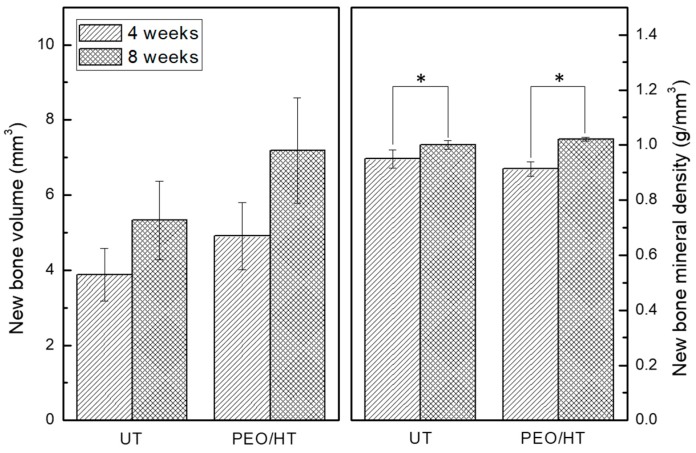
Quantitative volume and mineral density for the new bone formed in rat calvarial defects calculated from the Micro-CT data (* means *p* < 0.05).

**Figure 7 materials-12-02684-f007:**
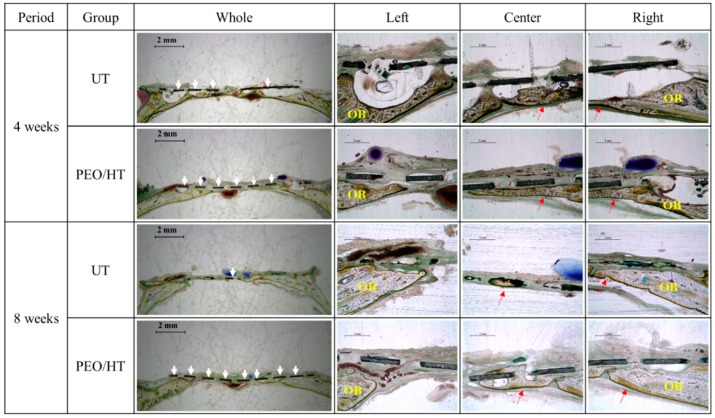
Histological images for bone regeneration in the rat calvarial defect after four weeks and eight weeks of implantation. (Red arrows = newly formed bone; yellow-OB = old bone; white arrows = magnesium mesh).

**Figure 8 materials-12-02684-f008:**
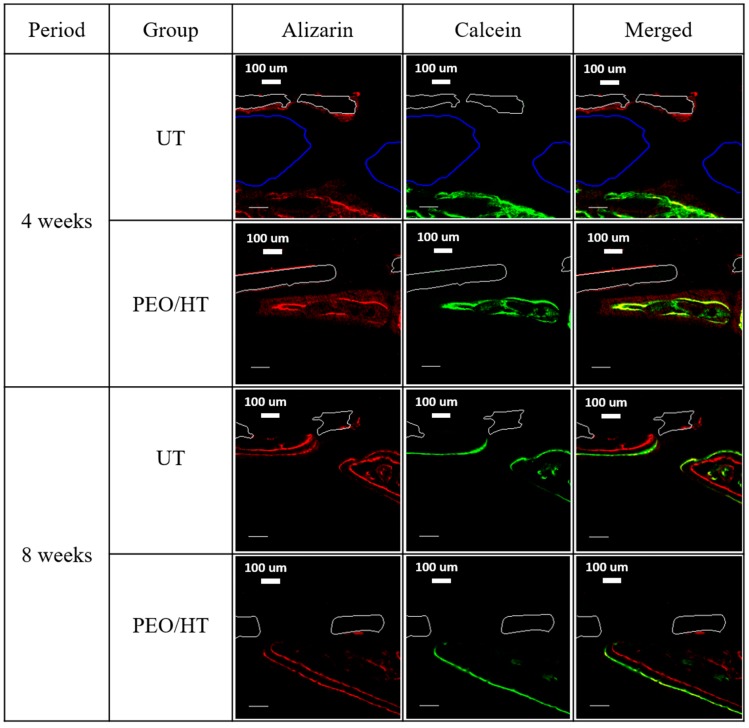
Fluorescent microphotograph after four weeks and eight weeks of implantation. The red label and green label represent the alizarin and calcein. The white solid outline represents the magnesium mesh, and the blue solid line represents the gas cavity.

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
