# Peer review of "Surface Modification of Pure Magnesium Mesh for Guided Bone Regeneration: In Vivo Evaluation of Rat Calvarial Defect"

_materials, 2019, doi:10.3390/ma12172684_

Round 1
Reviewer 1 Report
The authors presented a good work , but this manuscript presented many English problems. Profession English editing is suggested for this manuscript.
The other problems are listed as below :
Abstract: The research objectives are the specific accomplishments the researcher hopes to achieve by the study. Please revise the objective (also in introduction) and change it, is written as a description of what was done, instead of what wanted to be achieved by the study.
Line 38: …bony window or bony defect. Please add reference.
Line 36: Is it in the field of Periodontics or just oral implants?
Line 84: reference 26 investigated the effects of anodization‐cyclic precalcification‐heat (APH) treatment on the bonding ability of Ca‐P coating to the parent metal and osseointegration of Ti‐6Al‐7Nb implants; not Magnesium. Please remove
Line 90: Please do a compression test to determine the viability of Mg as a mesh.
Line 94: Figure 1 A: improve image quality, is too dark. For the measurements write the shaft = 4.9mm, other way it will look like the mesh’s diameter is not 10mm
Line 101: …according to ASTM G1-03. Please add reference.
Line 111: Surface characterization: How many samples were used for SEM, EDS, XRD? How were the samples prepared for these tests? There are also no SEM, EDS, XRD parameters given? Please indicate them for each test.
Line 129: Was Skyscan 1076 the machine or the software used to observe morphological changes of meshes during immersion test? Was a 3D or 2D observation?
Line 142: use same style as other headers.
Line 155: ad libitum, is a Latin term please write in italic.
Line 160: 2.5.2. Micro-computed tomography scanning (Micro-CT): describe how new bone volume and new bone mineral density were measured.
Line 168: 2.5.3. Histological analysis and fluorescent analysis. Please indicate the type of staining used for Histological analysis.
Line 172-182: Was this methodology done for the first time? Please indicate references where necessary.
Line 185: (both from Sigma)…write full name and country.
Figure 4: Indicate statistical significant difference between groups in figure B.
Line 244-246: Even though there was no statistically significant difference between groups, both bone volume and mineral density of PEO/HT mesh were higher than that of untreated Mg meshes at 8 weeks.
Due to the lack of statistically significant difference between new bone volume and new bone mineral density, is it possible for authors to do ‘Cortical defect closure measurements’ with Micro CT 3D results?
Line 286: Discussion: Indicate why is necessary to decrease the reabsorption rate of a mesh for GBR. Discussion is weak, more references ae necessary to support findings without statistically significant difference results.
Line 319: …which increased the corrosion resistance and promoted the bone formation in vivo. Indicate reference
Line 339: How do you know were cavities of hydrogen?
Author Response
We are very grateful for the reviewer to review our manuscript carefully and give valuable comments. We read your opinion very seriously and really honored to correct the article follow comments. We hope that this revised manuscript is acceptable for publication in Materials.
[Overall]
We carefully read through the reviewer’s comments to author and took specific solutions for each of them, and then modified necessary additional information to make this paper optimize in the revised manuscript according to the comments. The reviewer’s comments and our answers are provided as follows, and the modified contents were highlighted using yellow in the revised manuscript. It is highly appreciated for the review to make this paper better.
Comments of Reviewer 1:
1. The authors presented a good work, but this manuscript presented many English problems. Profession English editing is suggested for this manuscript.
[Answer] According to the comment, English language editing of our manuscript was conducted by EditageOnlineTM. Revisions of English language editing were not highlighted in manuscript due to lots of modification.
2. Abstract: The research objectives are the specific accomplishments the researcher hopes to achieve by the study. Please revise the objective (also in introduction) and change it, is written as a description of what was done, instead of what wanted to be achieved by the study.
[Answer] According to the comment, I had revised the contents of abstract and introduction as follows:
Abstract: Line 21-28
“In this study, the surface modification of pure magnesium mesh was conducted by plasma electrolytic oxidation and hydrothermal treatment to retard its biodegradation in physiological environment. And then weight loss measurement and surface characterization were performed after immersion test in simulated body fluid. The effect of surface modification of magnesium mesh on the guided bone regeneration was evaluated through in vivo test using rat calvarial defect model. It was identified that the biodegradation of Mg mesh was obviously retarded, and the volume and density of new bone formed in calvarial defect increased by the surface modification in comparison with them of the pristine Mg mesh.”
Changed into:
“In this study, surface modification of pure magnesium mesh was conducted by plasma electrolytic oxidation and hydrothermal treatment to form a densely protective layer on the Mg substrate. The protective layer mainly consisted of Mg(OH)2 with amorphous calcium phosphate. Then, weight loss measurement and Micro-CT imaging were performed after an immersion test in simulated body fluid. The effect of surface modification of magnesium mesh on the guided bone regeneration was evaluated through in vivo test using the rat calvarial defect model. The biodegradation of Magnesium mesh was identified to be significantly retarded. And the surface modification of Mg also can improve the bone volume and bone density of calvarial defect in comparison with them of the pristine Mg mesh.”
Introduction: Line 91-96
“In this study, the protective coating containing calcium and phosphorus on the surface of pure magnesium mesh was formed via a two-step treatment combined plasma electrolytic oxidation (PEO) and hydrothermal treatment (HT) to retard the initial degradation rate and to enhance biocompatibility. And then, the effect of surface modification on the corrosion behavior and bone regeneration capacity of magnesium mesh was investigated through immersion test and in-vivo test.”
Changed into:
“The current aim of this study was to evaluate the potential of surface-modified Mg as a biomedical material. The degradation of magnesium was investigated by weight loss measurement and Micro-CT imaging after an immersion test in simulated body fluid. The osteogenesis of magnesium was evaluated by in-vivo test in 8mm-critical size of rat calvarial model.”
3. Line 38: …bony window or bony defect. Please add reference
[Answer] I had add references as follows:
[2] Vertical Guided Bone Regeneration using Titanium-reinforced d-PTFE Membrane and Prehydrated Corticocancellous Bone Graft
[3] Healing of dehiscence‐type defects in implants placed together with different barrier membranes: a comparative clinical study
4. Line 36: Is it in the field of Periodontics or just oral implants?
[Answer] According to the comment, I had changed “Especially in oral implantology” into “Particularly in the surgery of maxillofacial”, this field included many kinds of surgery which required bone regeneration, for example, reconstruction of the bone structure after excision of ameloblastoma and jaw tumors, augmentation of deficient height of alveolar ridges caused by periodontics, and increase of jaw bone horizontally or vertically before implantation at site of tooth loss.
5. Line 84: reference 26 investigated the effects of anodization-cyclic precalcification-heat (APH) treatment on the bonding ability of Ca-P coating to the parent metal and osseointegration of Ti-6Al-7Nb implants; not Magnesium. Please remove
[Answer] According to the comment, I had removed the reference 26.
6. Line 90: Please do a compression test to determine the viability of Mg as a mesh.
[Answer] Thanks for your good comment. We found the ASTM international standard (ASTM E9-89a), which refers to compression testing of metallic materials at room temperature. In the part of the test specimen, the dimensions for testing of the thin sheet were depended on the type of compression jig, but the minimum of the specimen thickness is 0.15mm. However, the thickness of 0.1mm of the specimen of present study (as rolled, 100mm*100mm*0.1mm, Goodfellow, England) is not fit for compression test. For the existed materials, the compression test is difficult to accomplish. In future research, we will more focus on this characteristic of Mg mesh.
7.Line 94: Figure 1 A: improve image quality, is too dark. For the measurements write the shaft = 4.9mm, other way it will look like the mesh’s diameter is not 10mm
[Answer] According to the comment, I had changed the Figure 1 A at a more bright background. About the real size of the sample, I had made the images to show the dimension of mesh and the shaft.
8. Line 101: …according to ASTM G1-03. Please add reference.
[Answer] According to the comment, I had add ASTM international standard as a reference [30].
9. Line 111: Surface characterization: How many samples were used for SEM, EDS, XRD? How were the samples prepared for these tests? There are also no SEM, EDS, XRD parameters given? Please indicate them for each test.
[Answer] According to the comment, I had revised the contents into:
Line 111-117
“To observe the microstructure and the chemical composition of specimens, untreated Mg mesh, PEO-treated Mg mesh and PEO/HT-treated Mg mesh were identified through scanning electron microscopy microscope (SEM; JSM-5900, JEOL, Japan) in conjunction with energy dispersive spectroscopy (EDS; Bruker, Billerica, MA, USA). The cross-sectional analysis for the coating layer of PEO-treated Mg mesh and PEO/HT-treated Mg mesh was conducted using field emission scanning electron microscopy (FE-SEM, SU-70; Hitachi, Tokyo, Japan) after ion milling. Phase identification of untreated Mg mesh, PEO-treated Mg mesh and PEO/HT-treated Mg mesh were analyzed using X-ray diffractometer (X`pert Powder, PANalytical, Netherlands) with 2θ ranging from 10° to 90° by Cu-Kɑ(λ=0.154060 nm) at a step of 0.03°. All of the above tests, 3 samples per group were used.”
10. Line 129: Was Skyscan 1076 the machine or the software used to observe morphological changes of meshes during immersion test? Was a 3D or 2D observation?
[Answer] Skyscan 1076 is a 3D reconstruction machine under SkyScan Company. After Micro-CT scan, CTvox software (Ver 3.0, Skyscan) was used to reconstructed structure of sample.
11. Line 142: use same style as other headers.
[Answer] According to the comment, I had changed subtitle into “Surgical procedures”.
12. Line 155: ad libitum, is a Latin term please write in italic.
[Answer] According to the comment, I had revised “ad libitum” to "ad libitum”.
13. Line 160: 2.5.2. Micro-computed tomography scanning (Micro-CT): describe how new bone volume and new bone mineral density were measured.
[Answer] According to the comment, I had revised the contents and add the details about the calculated process as follows:
Line 160-166 had changed into:
“The dissected block specimens were immersed into 10% formalin in a plastic tube with label, then sent to the Center for University-Wide Research Facilities (CURF) at Chonbuk National University at the 4-week and 8-week time points. Dissected specimens were scanned using Micro-CT (Skyscan 1076, Skyscan, Aartselaar, Belgium) at 100 kV and 100 μA with the 680ms of exposure time, 0.6° of rotation intervals, and 360° of rotation angle, to obtain qualitative and quantitative data of bone regeneration. The 3D images reconstructed with NRecon software (Ver 1.7.0, Skyscan).
For the calvarial defect, a cylindrical region of interest (ROI) with a diameter of 8mm (the size of the cranial defect) and height about 3mm( included newly formed bone and magnesium mesh), was chosen for calculating the bone qualitative and quantitative results. The gray thresholds of osseous tissue were 110-255 in Hounsfield Units (HU). New bone volume (mm3) and new bone mineral density (g/mm3) were quantified using CTAn software (Skyscan). The 3D structure reconstructed by ®CTvoxs software (Ver 3.0, Skyscan). ”
14. Line 168: 2.5.3. Histological analysis and fluorescent analysis. Please indicate the type of staining used for Histological analysis.
[Answer] According to the comment, I had revised the contents and add the details about the calculated process as follows:
Line 171-173 had changed into:
“After Micro-CT scanning, the blocks were fixed in new 10% formalin for 1day and stained in Villanueva solution (Polysciences, Inc., Germany). Villanueva is a bone stain which is widely used for effective staining of mineralized or uncalcified bone, especially for the study of new bone formation in hydroxyapatite implants and bone grafts. Then, the blocks were dehydrated in ethanol with varying concentrations (80%, 90%, 95%, and 100%) and 100% acetone.”
15. Line 172-182: Was this methodology done for the first time? Please indicate references where necessary.
[Answer] This was not the first time to using this methodology for blocks staining and embedding. The method of staining with Villanueva and embedding with MMA has been used many times in our previous studies. Therefore I had revised Line 168 into "When finished Micro-CT scanning, the specimen began with a series of process for embedding and staining, following the methodology used in our previous study [31]"
[31] Enhancement of bone formation on LBL-coated Mg alloy depending on the different concentration of BMP-2
16. Line 185: (both from Sigma)…write full name and country.
[Answer] I had changed into “Sigma, St. Louis, MO, USA"
17. Figure 4: Indicate statistical significant difference between groups in figure B.
[Answer] It is highly appreciated for the reviewer’s valuable comment. According to your comment, I had checked the data once again and found some mistake, then I had revised Figure 4B and sentences in my manuscript, as follows.
Figure 4: details in attachment
Line 225-227 had changed into:
“Between the surface treated samples, the weight loss ratio (21.67 ± 4.56 %) of PEO mesh also had a significant difference with that (15.08 ± 2.89 %) of PEO/HT mesh. All the above, the surface-modified PEO / HT group significantly improved the biocorrosion resistance of pure Mg.”
18. Line 244-246: Even though there was no statistically significant difference between groups, both bone volume and mineral density of PEO/HT mesh were higher than that of untreated Mg meshes at 8 weeks.
Due to the lack of statistically significant difference between new bone volume and new bone mineral density, is it possible for authors to do ‘Cortical defect closure measurements’ with Micro CT 3D results?
[Answer] According to the comment, we had discussed the feasibility of cortical defect closure measurement of magnesium. There are certain difficulties in the calculation. Conventionally, the density of Mg was 1.74-2.0, which was similar to the density of bone(1.8-2.1). Mg, a biomedical application which has the ability of biodegradation and the osteogenic, results in that the density of Mg decreased and that of bone increased after implantation. Therefore, the density values overlapped to a large extent and had an effect on the measurement of bone formation and bone mineral density than nonresorbable Titanium(the density of 4.4-4.5). The new bone volume and bone mineral density values we currently calculate for magnesium are accurate values in a broad sense. The gray value setting of CTAn software includes lower density bone and higher density magnesium, which cannot be delimited like titanium and the cortical bone.
19. Line 286: Discussion: Indicate why is necessary to decrease the reabsorption rate of a mesh for GBR. Discussion is weak, more references are necessary to support findings without statistically significant difference results.
[Answer] According to the comment, I had revised the contents of discussion and add the content of necessarily of decrease the absorption rate of Mg mesh for GBR.
Discussion: Line 286-291
“A large number of researches have been performed to enhance the corrosion resistance of magnesium at the preliminary stage for biomedical application, since the high corrosion rate of pure magnesium or its alloys in physiological environment results in accumulation of hydrogen gas and ions of magnesium and alloy elements released from implants as well as increment of pH at tissue near implant, which can cause side effects like inflammatory response and hindrance of bone formation in addition to unpredictable fracture of implant[29].”
Changed into:
“Bioabsorbable barrier membranes are widely using in GBR of dentistry because the secondary surgery for removal of membrane is not required. However, one of shortcomings of absorbable membranes is the lack of rigidity, leading the collapse into wound and subsequent compromised volume or failure of bone regeneration. Even if successful in bone regeneration, the morphology often typically results in a rounded crestal ridge form, requiring extra grafting at the time of implantology.[32,33] Pristine Mg as biomedical material rapidly degraded in physiological environment, losing advantages of metal rigidity and space maintenance, indicating unable to provide necessary support until natural bone tissue complete healing. Meanwhile, the high corrosion rate of magnesium or its alloys at the preliminary stage of implantation resulted in gas accumulation and ions released as well as variable of pH at tissue around the implant, which can cause turbulent of the stable environment, interference with cell adhesion, and hindrance of bone formation [34,35]. Hence, a large number of researches have been performed to enhance the corrosion resistance and decrease the initial biodegraded rate of magnesium in biomedical application.”
[32] Schliephake H, Dard M, Planck H, Hierlemann H, Stern U. Alveolar ridge repair using resorbable membranes and autogenous bone particles with simultaneous placement of implants: An experimental pilot study in dogs. Int J Oral Maxillofac Implants 2000; 15: 364–373
[33] Hämmerle CH, Jung RE, Yaman D, Lang NP. Ridge augmentation by applying bioresorbable membranes and deproteinized bovine bone mineral: A report of twelve consecutive cases. Clin Oral Implants Res 2008; 19: 19–25.
[35] High purity biodegradable magnesium coating for implant application
20. Line 319: …which increased the corrosion resistance and promoted the bone formation in vivo. Indicate reference
[Answer] I had added the reference of [31] “Enhancement of bone formation on LBL-coated Mg alloy depending on the different concentration of BMP-2”
21. Line 339: How do you know were cavities of hydrogen?
[Answer] We highly appreciate the reviewer’s comment. According to the comment, all of "hydrogen gas" was replaced by "gas" in manuscript (5 in origin text: line 64, line 257, line 288, line338 and line341). Because this study did not perform the experiment of gas composition analysis, it is not appropriate to use "hydrogen", an accurate word, in the manuscript. Referring the prior study, I add some content of gas component analysis releasing by biomedical Mg in the discussion.
Discussion: Line 337-343
“At 8 weeks, most of the untreated mesh was degraded and the cavities of hydrogen gas was not observed any more. The dissolution of gas pocket might be because the corrosion rate of mesh decreased by the formation of byproducts such as Mg(OH)2 and MgxCay(PO4)z(OH)n complex compounds formed on the surface of mesh and the reduction of surface area where hydrogen gas can be evolved. However, tiny gas pockets were observed in new bone formed at the center of PEO/HT mesh, which resulted from the localized corrosion of mesh by destruction of PEO/HT coating.”
Changed into:
“At 8 weeks, most of the untreated mesh was degraded, and the cavities of gas were not observed anymore. The diminishment of gas generation might be due to the decrease in the corrosion rate of the mesh by the formation of byproducts, such as Mg(OH)2 and MgxCay(PO4)z(OH)n complex compounds, and the reduction in the surface area where gas can evolve. The disappeared of gas pockets indicated that the generation rate of H2 falls below the diffusion rate of H2[49]. However, tiny gas pockets were observed in the new bone formed at the center of the PEO/HT mesh, which resulted from the localized corrosion of mesh by the destruction of the PEO/HT coating. It was identified that the main component of gas released by Mg was hydrogen, and CO2 and CO constituted the residual gas[50]. When the Mg placed in body fluid environment that the hydrogen released rapidly (Mg + 2H2O → Mg(OH)2 + H2).”
[49] Fast escape of hydrogen from gas cavities around corroding magnesium implants
[50] Gas formation and biological effects of biodegradable magenesium in a preclinical and clinical observation

Reviewer 2 Report
The entitled “Surface modification of the pure magnesium mesh for guided bone regeneration: in vivo evaluation using rat calvarial defect” by Wu et al. investigates modification of pure magnesium mesh using plasma electrolytic oxidation and hydrothermal treatment on the mesh biodegradation in vitro and guided bone regeneration in vivo. It was concluded that the biodegradation of terated Mg mesh was retarded, and the volume and density of new bone formed in calvarial defect increased by the surface modification in comparison with the untreated Mg mesh. The question of the article is clear and well investigated, however there is weakness in the data interpretation, discussion and aticles language needs to be improved preferably edited by native speaker. I would like to make a few comments in regards.
Is the sample size in the in vivo experiment enogh for reliable results. The authors concluded in the abstarct that the mg treated mech induced more bone formation and density based on results which is not statisticaly significant. The non-significant differences results either because of small sample size or the effect of the change is not significant. Accoding to the statistics of the in vivo data, the authors can not say that the surface treatment lead to higer bone formation. The authors claim that the spaces created under the untreated mesh is hydrogen gas. How was that investigated. Clarify in the results text The authors have mentioned that Red-Alizarin complexion and green-Calcein were used to evaluate the dynamics of bone formation, but they have not clarified which tissues/ cells / structures that they are targeting and why they judge that bone formation and remodeling occur simultaneously at four week of healing. Does the blue line which represent the gas cavity in the Fig 7 resulted from any stain or? If any stain was used for this it should be mentioned. In the discussion the authors presented assumptions about the indirect effect of the mg corrosion on the bone healing. These assumptions should be investigated or at least supported by citing relevant previous studies. The authors has not provided clear disscusion on the plausible direct effect of the chemically modefied surface on the healing process. How the participated compunds Mg O, Mg (OH)2on the mesh surface could improve the bone regeneration. Were Ca snd P formed in form of amorphous cacium phosphate commound or in other forms?. How does this play role in bone healing. Labeling/ identification for structures and tissues in the fig 5 and fig 7 should be provided, i.e Identify for example the new bone, the old bone, the mech ets
Author Response
We are very grateful for the reviewer to review our manuscript carefully and give valuable comments. We read your opinion very seriously and really honored to correct the article follow comments. We hope that this revised manuscript is acceptable for publication in Materials.
[Overall]
We carefully read through reviewer’s comments to author and took specific solutions for each of them, and then modified necessary additional information to make this paper optimize in the revised manuscript according to the comments. The reviewer’s comments and our answers are provided as follows, and the modified contents were highlighted using yellow in the revised manuscript. It is highly appreciated for the review to make this paper better.
Comments of Reviewer 2:
1. The question of the article is clear and well investigated, however there is weakness in the data interpretation, discussion and article’s language needs to be improved preferably edited by native speaker.
[Answer] According to the comment, English language editing of our manuscript was conducted by EditageOnlineTM. Revisions of English language editing were not highlighted in manuscript due to lots of modification. In terms of data interpretation, it has been supplemented and revised based on comments.
2. Is the sample size in the in vivo experiment enough for reliable results?
[Answer] According to the reference, the GPower software (Ver 3.1.9.4) was used to calculated the sample size.
The results are as follows: details in attachment
The results showed that 16 of the 2 groups is enough for this study. The effect size of 0.8 belongs to the "large" in general. However, it belongs to the "medium" degree in work with laboratory animals. The above shows that there will be statistical differences when the medium or large effect is shown in the experiment.
Reference:
[1] Jaykaran Charan and N. D. Kantharia,How to calculate sample size in animal studies?
[2] Cohen J. 2nd ed. Hillsdale, NJ: Lawrence Erlbaum; 1988. Statistical Power Analysis for the Behavioral Sciences.
[3] Software G Power (Faul, Erdfelder, Lang and Buchner, 2007)
3. The authors concluded in the abstract that the mg treated mesh induced more bone formation and density based on results which is not statistically significant. The non-significant differences results either because of small sample size or the effect of the change is not significant. According to the statistics of the in vivo data, the authors cannot say that the surface treatment lead to higher bone formation.
[Answer] According to the comment, I had revised the contents of abstract as follows:
Abstract: Line 27-28 Changed into:
“The biodegradation of Magnesium mesh was identified to be significantly retarded. And the surface modification of Mg also can improve the bone volume and bone density of calvarial defect in comparison with them of the pristine Mg mesh.”
4. The author’s claim that the spaces created under the untreated mesh is hydrogen gas. How was that investigated? Clarify in the results text.
[Answer] We highly appreciate the reviewer’s comment. According to the comment, all of "hydrogen gas" was replaced by "gas" in manuscript (5 in origin text: line 64, line 257, line 288, line338 and line341). Because this study did not perform the experiment of gas composition analysis, it is not appropriate to use "hydrogen", an accurate word, in the manuscript. Referring the prior study, I add some content of gas component analysis releasing by biomedical Mg in the discussion.
Discussion: Line 337-343
“At 8 weeks, most of the untreated mesh was degraded and the cavities of hydrogen gas was not observed any more. The dissolution of gas pocket might be because the corrosion rate of mesh decreased by the formation of byproducts such as Mg(OH)2 and MgxCay(PO4)z(OH)n complex compounds formed on the surface of mesh and the reduction of surface area where hydrogen gas can be evolved. However, tiny gas pockets were observed in new bone formed at the center of PEO/HT mesh, which resulted from the localized corrosion of mesh by destruction of PEO/HT coating.”
Changed into:
“At 8 weeks, most of the untreated mesh was degraded, and the cavities of gas were not observed anymore. The diminish of gas generation might be due to the decrease in the corrosion rate of the mesh by the formation of byproducts, such as Mg(OH)2 and MgxCay(PO4)z(OH)n complex compounds, and the reduction in the surface area where gas can evolve. The disappeared of gas pockets indicated that the generation rate of H2 falls below the diffusion rate of H2[49]. However, tiny gas pockets were observed in the new bone formed at the center of the PEO/HT mesh, which resulted from the localized corrosion of mesh by the destruction of the PEO/HT coating. It was identified that the main component of gas released by Mg was hydrogen, and CO2 and CO constituted the residual gas[50]. When the Mg placed in body fluid environment that the hydrogen released rapidly (Mg + 2H2O → Mg(OH)2 + H2).
[49] Fast escape of hydrogen from gas cavities around corroding magnesium implants
[50] Gas formation and biological effects of biodegradable magnesium in a preclinical and clinical observation
5. The authors have mentioned that Red-Alizarin complexion and green-Calcein were used to evaluate the dynamics of bone formation, but they have not clarified which tissues/ cells / structures that they are targeting and why they judge that bone formation and remodeling occur simultaneously at four week of healing.
[Answer] According to the comment, I had revised the contents as follows:
1) Which tissues/ cells / structures that they are targeting
Line 266-269 had added the contents:
“To clarify the new bone formation dynamics in the defect beneath the magnesium mesh, the sequential fluorochrome labels were administered subcutaneously for each group at the different time. The sequence of new bone formation was confirmed by red label of alizarin complexone and the green-label of calcein.”
Changed into:
“To clarify the new bone formation dynamics in the defect beneath the magnesium mesh, the sequential fluorochrome labels were administered subcutaneously for each group at the different time. The sequence of new bone formation was confirmed by red label of alizarin complexone and the green-label of calcein. After administration, fluorochrome labels, bound to Ca ions of the body, were incorporated in the form of HA at the sites of mineralization. This means that the fluorescent label delimited the mineralization front at the time of administration. As a result, bone formation can be tracked by administrating different labels at specific time intervals.”
2) Why they judge that bone formation and remodeling occur simultaneously at four week of healing.
The 4-weeks sacrificed group received the red-alizarin post-surgery and the calcein on 2 weeks. According to the working mechanism of fluorochrome, the green in red label indicated the bone mineralization site after 2 weeks is located inside the previously formed bone, changing the density of bone or remolding the structure. Hence I revised the sentence:
Line 270-273 had been revised:
“In the case of rats sacrificed at 4 weeks after administering the alizarin and the calcein at right after operation and 2 weeks, the green label of calcein appeared inside the red label of alizarin or two labels overlapped in both groups, which indicates that formation and remodeling of new bone layer were occurred simultaneously.”
Changed into:
“In the case of rats sacrificed at 4 weeks received the red-alizarin post-surgery and the calcein on 2 weeks. According to the working mechanism of fluorochrome label, the green label of calcein appeared inside the red label of alizarin or two labels overlapped in both groups, which indicates that deformation/remodeling of new bone layer occurred.”
6. Does the blue line which represent the gas cavity in the Fig 7 resulted from any stain or? If any stain was used for this it should be mentioned
[Answer] Thanks for your good comment. Refer the blue line, I was confused of Figure 7 or Figure 8. Hence I divided to two parts.
In Figure 7:
The blue line in the image of center points of 8-weeks UT group. It is the bioabsorbable silk (5-0 poly-glactin 910) used to suture the periosteum for fixation of Mg mesh. After the stain of the purple Vallianueva and a series of graded dehydrated treatment, the color changed from transparent, purple, and fade to blue.
In Figure 8: details in attachment
The blue line in the image of 4-weeks UT group. Blue solid-lines, manual drawing, represent the gas cavity generated by biodegradable untreated Mg mesh. It observed by confocal laser scanning microscope (CLSM 510 Meta, Zeiss, Germany), and analyzed by confocal microscopy software (Ver 4.2.0., Zeiss LSM Data Server). The merged images of white channel observed the gas cavity located at the site between new-formed bone and biodegrading Mg mesh. In red-alizarin and green-calcein overlapping positions, the black channel displayed more obvious and clear than that of the white channel. Hence the merged images of black were chosen in this manuscript.
7. In the discussion the authors presented assumptions about the indirect effect of the mg corrosion on the bone healing. These assumptions should be investigated or at least supported by citing relevant previous studies.
[Answer] According to the comment, I had revised the contents as follows:
Discussion: Line 286-291
“A large number of researches have been performed to enhance the corrosion resistance of magnesium at the preliminary stage for biomedical application, since the high corrosion rate of pure magnesium or its alloys in physiological environment results in accumulation of hydrogen gas and ions of magnesium and alloy elements released from implants as well as increment of pH at tissue near implant, which can cause side effects like inflammatory response and hindrance of bone formation in addition to unpredictable fracture of implant[29].”
Changed into:
“Pristine Mg as biomedical material rapidly degraded in physiological environment, losing advantages of metal rigidity and space maintenance, indicating unable to provide necessary support until natural bone tissue complete healing. Meanwhile, the high corrosion rate of magnesium or its alloys at the preliminary stage of implantation resulted in gas accumulation and ions released as well as variable of pH at tissue around the implant, which can cause turbulent of the stable environment, interference with cell adhesion, and hindrance of bone formation [34,35]. Hence, a large number of researches have been performed to enhance the corrosion resistance and decrease the initial biodegraded rate of magnesium in biomedical application.”
[35] High purity biodegradable magnesium coating for implant application
8. The authors has not provided clear discussion on the plausible direct effect of the chemically modified surface on the healing process. How the participated compounds Mg O, Mg (OH)2 on the mesh surface could improve the bone regeneration. Were Ca and P formed in form of amorphous calcium phosphate compound or in other forms? How does this play role in bone healing.
[Answer] It is highly appreciated for the reviewer’s valuable comment. I had known what is missing in my manuscript. Therefore, I added the analysis of the chemically modified PEO / HT based on the comments.
Discussion: Line 318-322
“In present study, even though Ca and P were detected by EDS analysis, Mg(OH)2 was mainly formed on Mg mesh without the formation of crystalline calcium phosphate phase when the hydrothermal treatment was conducted after PEO coating including P as confirmed by XRD analysis, this Mg(OH)2 layer with about 10μm retarded the degradation of Mg mesh obviously as confirmed by immersion test.
Changed into:
“In the present study, Mg(OH)2 was the main crystalline phase forming on PEO/HT treated Mg mesh, which retarded the degradation of Mg mesh significantly and enhanced new bone formation, as confirmed by immersion test and in vivo test. C. Janning et al [43] proved that bone growth enhancement around the slowly dissolving Mg(OH)2 cylinder in rabbit model caused by enhancing osteoblast activity and decreasing bone resorption. According to the EDS and XRD data of stated experiment, the Ca/P ratio of the specimens was less than 1.0, which suggests that the precipitated calcium phosphate is amorphous. However, this didn't imply the absence of microcrystalline phase. The mineral phase of hard tissue has an apatite structure consisting of non-stoichiometric HAP and trace element like Mg2+, CO32-, Cl-[44,45]. When the Mg concentration is high, the apatite exists in the form of (Ca1-xMgx)10(PO4)6OH2, which has biocompatibility to MG63 cells. These are consistent with the presented results of the animal test, the protective coating layer containing Ca-P of PEO/HT has a good affinity of bone-implant interface, high corrosion resistance and increase the bone formation at the initial implantation.”
[43] Magnesium hydroxide temporarily enhancing osteoblast activity and decreasing the osteoclast number in peri-implant bone remodeling
[44] Synergistic effects of magnesium and carbonate on properties of biological and synthetic apatites
[45] Osteoblastic cell response on magnesium-incorporated apatite coating: The first international symposium on surfaces and interfaces of biomaterials.
9. Labeling/ identification for structures and tissues in the fig 5 and fig 7 should be provided, i.e. Identify for example the new bone, the old bone, the mesh ets
[Answer] According to the comment, I had revised the figures.

Round 2
Reviewer 1 Report
The authors revised the manuscript by the reviewers' opinions. I suggest to accept this manuscript for publication .